# Structure Elucidation of Triterpenoid Saponins Found in an Immunoadjuvant Preparation of *Quillaja brasiliensis* Using Mass Spectrometry and ^1^H and ^13^C NMR Spectroscopy

**DOI:** 10.3390/molecules27082402

**Published:** 2022-04-08

**Authors:** Federico Wallace, Carolina Fontana, Fernando Ferreira, Cristina Olivaro

**Affiliations:** 1Espacio de Ciencia y Tecnología Química, CENUR Noreste, Universidad de la República, Tacuarembó 45000, Uruguay; federico.wallace@cut.edu.uy (F.W.); ff@fq.edu.uy (F.F.); 2Laboratorio de Espectroscopía y Fisicoquímica Orgánica, Departamento de Química del Litoral, CENUR Litoral Norte, Universidad de la República, Paysandú 60000, Uruguay; 3Laboratorio de Carbohidratos y Glicoconjugados, Departamento de Química Orgánica, Facultad de Química, Universidad de la República, Montevideo 10800, Uruguay

**Keywords:** *Quillaja brasiliensis*, immunoadjuvant saponins, structural analysis, NMR spectroscopy, QS-21 isomer

## Abstract

An immunoadjuvant preparation (named Fraction B) was obtained from the aqueous extract of *Quillaja brasiliensis* leaves, and further fractionated by consecutive separations with silica flash MPLC and reverse phase HPLC. Two compounds were isolated, and their structures elucidated using a combination of NMR spectroscopy and mass spectrometry. One of these compounds is a previously undescribed triterpene saponin (**Qb1**), which is an isomer of **QS-21**, the unique adjuvant saponin employed in human vaccines. The other compound is a triterpene saponin previously isolated from *Quillaja saponaria* bark, known as **S13**. The structure of **Qb1** consists of a quillaic acid residue substituted with a β-d-Gal*p*-(1→2)-[β-d-Xyl*p*-(1→3)]-β-d-Glc*p*A trisaccharide at C3, and a β-d-Xyl*p*-(1→4)-α-l-Rha*p*-(1→2)-[α-l-Ara*p*-(1→3)]-β-d-Fuc*p* moiety at C28. The oligosaccharide at C28 was further substituted at O4 of the fucosyl residue with an acyl group capped with a β-d-Xyl*p* residue.

## 1. Introduction

*Quillaja brasiliensis* (A. St.-Hill. & Tul.) Mart. (*Quillajaceae*) (*Qb*) is an endemic tree species of South America. It belongs to a very restricted botanical family that includes a single genus (*Quillaja Molina*) with only two currently accepted species, with the other one being *Quillaja saponaria* Molina (*Qs*) [1].

Saponins from *Quillaja* plants present similar chemical and biological properties, being the immunoadjuvant and immunoestimulant activities the most relevant. These saponins, either alone or in colloidal formulations, have proved effective to generate both humoral and cellular response against the co-administered antigens, thus becoming important compounds for vaccine development [2,3,4,5,6,7,8,9].

*Qs* saponins are employed in the manufacture of vaccines for human and veterinary use, and the bark of this tree is one of the main sources of triterpene saponins worldwide [10,11]. Quil-A^®^, a commercial mixture of various saponins obtained from the bark extract of *Qs*, is used in veterinary vaccines, but it has not been considered for inclusion in human formulations due to its high reactogenicity. However, **QS-21**, a mixture of two isomeric saponins (**QS-21Xyl** and **QS-21Api**) that are present in Quil-A^®^ (Figure 1), has been shown to be less reactogenic than the latter while maintaining adjuvant properties. It has been tested as adjuvant in human vaccines either for the prevention or treatment of diseases such as cancer, HIV, tuberculosis, Alzheimer’s, and COVID-19 [12,13,14,15]. Currently, there are two licensed vaccines for human use that contain the AS01 adjuvant, which is a combination of **QS-21** with monophosphoryl lipid A and liposomes: MosqirixTM, a malaria vaccine approved in 2015 for use in children living in areas where this disease is endemic, and ShingrixTM, an herpes zoster vaccine approved in 2018 [16,17]. It is worth pointing out that **QS-21** induces a strong Th1/Th2 immune response with cytotoxic T lymphocytes (CTL) production in a relatively short time, as compared with other adjuvants [18]. 

Even though the study of *Qb* saponins started relatively later, *Qb* represents a natural renewable alternative source of these products, since saponins are also abundant in the leaves. It has been previously shown that the aqueous extract and some purified fractions obtained from leaves of *Qb* have immunoadjuvant activity comparable to that of Quil-A^®^ [19,20]. The adjuvant potential of *Qb* saponins has been confirmed in experimental vaccines against different viruses in murine models. These studies have been performed with the aqueous extract, purified fractions, and nanoparticles derived from purified fractions, which have been formulated with and without the presence of antigen [2,3,19,20,21,22,23,24]. Recently, we presented the first structural studies of an immunoadjuvant fraction of saponins obtained from leaves, named Fraction B (FB), using direct infusion and liquid chromatography/electrospray ionization ion trap multiple stage mass spectrometry (DI-ESI-IT-MS^n^ and LC-ESI-IT-MS^2^) in combination with classical methods of monosaccharide and methylation analysis [25,26]. Forty-eight bidesmosidic saponins, bearing five types of triterpenic aglycones including quillaic acid, hydroxylated quillaic acid (22β), gypsogenin, phytolaccinic acid and its O-23 acetate, were preliminarily characterized. In a continuation of the characterization of saponins from FB, we performed the isolation and structure elucidation of one undescribed triterpene saponin, which is isomer of **QS-21**, named **Qb1**, and other saponin previously reported in *Qs*, known as **S13** [27]. Their structures were elucidated using a combination of mass spectrometry (ESI-MS) and NMR spectroscopy methods.

## 2. Results and Discussion

### 2.1. Saponin Qb1

Three enriched saponins fractions (B1, B2, and B3) were collected after chromatographic separation of FB [25,26] on a silica flash medium-pressure liquid chromatography (MPLC) column. Fraction B3 was further purified by semi-preparative-high performance liquid chromatography (HPLC) on a reverse phase column, affording a previously undescribed triterpenic saponin named **Qb1**. This compound was analyzed by liquid chromatography coupled to mass spectrometry (LC-MS) and showed a chromatographic peak with a retention time of 24.8 min, while *Qs* saponin **QS-21** showed a peak at 28.1 min, demonstrating that the commercial standard of **QS-21** used herein is not a mixture of the two isomers (QS-21Xyl and QS-21Api) but one of them. The multiplicity-edited ^1^H,^13^C-HSQC spectrum of **QS-21** showed in the region of the anomeric resonances seven cross-peaks corresponding to pyranosyl residues (δ_H_/δ_C_ 5.37/94.9, 5.18/100.8, 4.79/103.8, 4.61/104.9, 4.52/105.7, 4.47/107.1 and 4.38/104.5) and one cross-peak corresponding to a furanosyl residue (δ_H_/δ_C_ 4.98/108.7); the latter is in accordance with the presence of a α-l-Ara*f* residue [28]. The lack of another residue in the form of furanose (i.e., the anomeric carbon resonance of the β-d-Api*f* residue of QS-21Api isomer is expected at δ_C_~112 ppm) implies that the commercial standard contains **QS-21Xyl** as a major component. **Qb1** and **QS-21Xyl** exhibited not only the same molecular mass but also remarkable similar MS^2^ spectra (Figure 2B,C, respectively), obtained using identical experimental conditions. In a preliminary study [25], this saponin (**Qb1**) was characterized using LC-MS and corresponded to the saponin **14** mentioned in the original work. The MS spectrum of **Qb1** showed a deprotonated pseudomolecular ion at *m*/*z* 1988.0 [M-H]^−^. A general structure of **Qb1** is depicted in Figure 2A, which takes into consideration the most conserved structural features of *Qs* saponins reported previously. The MS^2^ spectra of the precursor ions [M-H]^−^ of **Qb1** and **QS-21Xyl** are shown in Figure 2B,C, respectively. In the case of **Qb1**, the daughter ion *m*/*z* 955.6 (fragment a) is consistent with a saponin composed of a quillaic acid residue substituted with a trisaccharide moiety at C3, in which X_0_ is a pentose residue. Furthermore, the daughter ion *m*/*z* 1511.7 (fragment b) indicates that the α-l-Rha*p*-(1→2)-β-d-Fuc*p* moiety is substituted with two pentose residues, but their exact location could not be determined using solely MS spectrometry (all the possible substitution positions are indicated as X_1_, X_2_, X_3_ and X_4_). Finally, the daughter ions *m*/*z* 1553.6 and 1725.8 (fragments c and d, respectively) are consistent with fragmentations of the Fa(I) and Fa(II) acyl chains, whereas the pseudomolecular ion *m*/*z* 1988.0 reveals that the acyl chain is capped with a pentose residue (X_5_). The above data are for **Qb1** (Figure 2B), and a similar analysis can be carried out for **QS-21Xyl** (Figure 2C).

A combination of 1D and 2D NMR experiments, such as ^1^H,^1^H-TOCSY, ^1^H,^1^H-NOESY, multiplicity-edited ^1^H,^13^C-HSQC, ^1^H,^13^C-H2BC, ^1^H,^13^C-HSQC-TOCSY, ^1^H,^13^C-HSQC-NOESY and ^1^H,^13^C-HMBC, were performed and permitted the total assignment of all the ^1^H and ^13^C signals of **Qb1** in methanol-*d4* solution. The proton spin systems were determined by analysis of ^1^H,^1^H-TOCSY spectra with increasing mixing times, using the anomeric proton signals as a starting point for the assignments. In some cases, ^1^H,^1^H-NOESY spectra were also used to establish intra-residue connectivities in residues with the *galacto*-configuration (i.e., assignment of H5 protons). The ^13^C signals were assigned using one-bond proton–carbon correlations from multiplicity-edited ^1^H,^13^C-HSQC spectra and two-bond heteronuclear correlations from the ^1^H,^13^C-H2BC spectra. The assignments carried out using homonuclear experiments (^1^H,^1^H-TOCSY and ^1^H,^1^H-NOESY) were also confirmed using ^1^H,^13^C-HSQC-TOCSY and ^1^H,^13^C-HSQC-NOESY spectra. The anomeric configurations of the sugar residues were established using ^3^*J*_H1,H2_ values measured directly from the ^1^H NMR spectrum anomeric proton signals [29,30] or ^1^*J*_C1,H1_ values measured from the residual coupled C1/H1 crosspeaks in the ^1^H,^13^C-HMBC spectra. The inter-residue correlations were determined using ^1^H,^1^H-NOESY and/or ^1^H,^13^C-HMBC spectra.

Quillaic acid was identified as the triterpene aglycone of the saponin by NMR spectroscopic data (Appendix A) and by comparison with literature data [28,31]. The δ_C_ values observed for C3 (86.4) and C28 (177.3) were also consistent with the bidesmosidic nature of this saponin and with our previous preliminary characterization by mass spectrometry [26,27].

The multiplicity-edited ^1^H,^13^C-HSQC spectrum of **Qb1** revealed eight resonances in the anomeric region (Figure 3B). The anomeric protons resonated as doublets at δ 5.41 (*J* = 8.1 Hz), 5.29 (*J* = 1.9 Hz), 4.80 (*J* = 7.1 Hz), 4.59 (*J* = 7.7 Hz), 4.47 (*J* = 7.7 Hz), 4.43 (*J* = 8.6 Hz), 4.41 (*J* = 6.7 Hz), and 4.27 (*J* = 7.7 Hz) in the ^1^H NMR spectrum. In the multiplicity-edited ^1^H,^13^C HSQC spectrum, these protons correlated to the carbon signals at δ_C_ 95.1, 101.8, 103.8, 104.9, 107.0, 104.6, 105.6 and 104.2, respectively. 

In the ^1^H,^1^H-TOCSY spectrum recorded with the longest mixing time (Appendix A), the residues with the anomeric resonances at 5.41 and 4.80 ppm showed correlations from H1 to H2–H4, revealing that they have the *galacto*-configuration (i.e., the correlations from H1 to H5 and H6 are not observed due to the small ^3^*J*_H4,H5_ value). Both monosaccharides showed intra-residue ^1^H,^1^H-NOESY correlations between H1 and H5, as well as large ^3^*J*_H1,H2_ values (8.1 and 7.1 Hz, respectively), indicating that they are found in a β-pyranose form. In the former monosaccharide, two intra-residue correlations could be observed in the ^1^H,^1^H-NOESY spectrum from H4 (5.29 ppm) to H5 (3.85 ppm) and H6 (1.08 ppm); the chemical shift of H6 revealed that this is a 6-deoxyhexose, and thus it is the β-d-Fuc*p* residue. The other monosaccharide showed a ^1^H,^1^H-TOCSY correlation from H5 to the H6a and H6b protons (3.73 and 3.76 ppm, respectively) and thus it can be assigned to the β-d-Gal*p* residue. The H1 resonance at 5.29 ppm showed a single correlation to H2 (3.96 ppm) in the aforementioned ^1^H, ^1^H-TOCSY spectrum, revealing that the monosaccharide has the *manno*-configuration (i.e., the small ^3^*J*_H1, H2_ and medium ^3^*J*_H2, H3_ values hamper the magnetization transference from H1 to protons beyond H2); however, the remaining protons in this spin system (H3–H6) could be assigned using the correlations from H2 observed in the same spectrum. In this case, the low chemical shift of H6 (1.30 ppm) is consistent with a 6-deoxyhexose, and thus this is the α-l-Rha*p* residue. In addition, the chemical shifts of this residue are remarkably similar to those reported previously for the 4-*O*-substituted α-l-Rha*p* residue of **QS-21Xyl** [30]. The ^1^*J*_C1, H1_ value (172 Hz), determined from the residual coupled C1/H1 crosspeak in the ^1^H,^13^C-HMBC spectrum, is consistent with this monosaccharide adopting an α-pyranose configuration [32]. The monosaccharide with H1 at 4.43 ppm showed ^1^H,^1^H-TOCSY correlations from the anomeric proton to H2-H5, revealing that this monosaccharide has a *gluco*-configuration. The large H1–H2 coupling constant (*J* = 8.6 Hz) is then consistent with a β-pyranose form and thus this residue can be assigned to the β-d-Glc*p*A. According to the MS data (Figure 1A), the remaining residues should all be pentoses. In the ^1^H, ^1^H-TOCSY spectrum (Appendix A), the residues with the anomeric protons at 4.59, 4.47, and 4.27 ppm showed patterns consistent with Xyl*p* residues (i.e., all protons correlations from H1 to H2–H5 could be traced in the spectrum recorded with τ_m_ 100 ms). In all three cases, intra-residue correlations from C5 to H1 were observed in the ^1^H,^13^C-HMBC spectrum, confirming that the monosaccharides are in the pyranose form; the large ^3^*J*_H1,H2_ values (7.7 Hz) indicate that these are all β-d-Xyl*p* residues. Finally, in the residue with the anomeric proton resonance at 4.41 ppm, ^1^H,^1^H-TOCSY correlations from H1 to H2–H4 could be identified, but no correlations were observed from H1 to H5_eq_ in the spectrum recorded with the longest mixing time (Appendix A); the H4 proton of this residue shows a sharp resonance, comparable to that of H4 of Fuc*p* and Gal*p*, which is consistent with this pentose being an Ara*p* residue (i.e., it is expected that ^3^*J*_H3,H4_, ^3^*J*_H4,H6ax_, and ^3^*J*_H4,H6eq_ have medium to small values [33]). Furthermore, key correlations observed in the ^1^H,^1^H-NOESY spectrum between H1-H5, and three-bond heteronuclear correlations observed in the ^1^H,^13^C-HMBC spectra from C5 to H1, confirmed that this residue is indeed adopting a pyranose form; based on the value of ^3^*J*_H1,H2_ (6.7 Hz), this residue can be assigned to an α-l-Ara*p*. It is worth pointing out that, in the saponins of *Q. saponaria* reported previously, the arabinose residues have always been found in furanose forms.

The ^1^H and ^13^C chemical shifts of the two oligosaccharide chains from **Qb1** are compiled in Table 1. All the monosaccharides identified herein are consistent with the monosaccharide analysis performed previously on FB [26]; the absolute configuration was assumed according to the saponins of the related species *Q. saponaria*. The substitution positions were deduced based on the high downfield chemical shifts at the substitution positions, in comparison to the respective free monosaccharides; thus, the C2, C3, and C4 chemical shifts of the Fuc residue (75.1, 81.9, and 74.7 ppm, respectively) are consistent with a trisubstituted monosaccharide →2,3,4)-β-d-Fuc*p*-(1→, the C2 and C3 chemical shifts of the GlcA residue (78.3 and 86.7 ppm, respectively) reveal a two-substituted monosaccharide →2,3)-β-d-Glc*p*A-(1→, and C4 of the Rha residue (84.1 ppm) indicate that this monosaccharide is →4)-α-l-Rhap-(1→. All pentoses, as well as the galactose residue, are expected to be unsubstituted, since no significant glycosylations shifts were observed apart from C1. The sequence of the two oligosaccharide chains and their connection to C3 and C28 of the aglycone were obtained from ^1^H,^13^C-HMBC and ^1^H,^1^H-NOESY experiments (Table 1). The oligosaccharide linked to C3 of the quillaic acid moiety was characterized as β-d-Gal*p*-(1→2)-[β-d-Xyl*p*-(1→3)]-β-d-Glc*p*A, since three-bond heteronuclear correlations were observed in the ^1^H,^13^C-HMBC spectrum from the anomeric protons of GlcA, Gal and Xyl(I) to the respective substitution positions (i.e., C3 of the quillaic acid moiety, and C2 and C3 of the GlcA residue, respectively). The chemical shifts of this trisaccharide moiety (Table 1) are consistent with those reported in bibliography for *Q. saponaria* saponins that share the same structural element [30,33]. In addition, the quillaic acid plus this trisaccharide were consistent with the MS^2^ spectrum, which showed an ion at *m*/*z* 955.6 (Figure 2A). The ^1^H,^13^C-HMBC spectrum showed a cross peak at δ 5.41 (H1 of Fuc)/177.3 (C28 of Qa) confirming that Fuc residue is linked to C28 of the aglycone. Correlations observed in the ^1^H,^13^C-HMBC spectrum (Appendix A) from the anomeric protons of the Rha and Ara residues to C2 and C3 of the fucosyl residue, and from H1 of Xyl(II) to C4 of the Rha residue are consistent with the following tetrasaccharide structure: β-d-Xyl*p*-(1→4)-α-l-Rha*p*-(1→2)-[α-l-Ara*p*-(1→3)]-β-d-Fuc*p*. All these data are also consistent with the correlations observed in the same spectrum from C1 of the aforementioned monosaccharides to the respective protons at the substitution positions, as well as those observed in the ^1^H,^1^H-NOESY spectrum (Table 1). The β-d-Fuc*p* residue is also substituted at O4 with a dimeric C9 acyl group capped with Xyl(III); the latter was demonstrated from the ^1^H,^13^C-HMBC correlation from H4 of the fucosyl residue to the carbon signal at δ_C_ 172.9 (C1 of the acyl group Fa(I)). All ^1^H and ^13^C signals for the acyl group were assigned (Appendix A) and compared with bibliographic data of *Q. saponaria* saponins [28,31], resulting in a 3,5-dyhidroxy-6-methyloctanoic acid moiety. Unlike the *Q. saponaria* saponins (that have an α-l-Ara*f* residue attached to this acyl group) (Figure 1), saponin **Qb1** has a β-d-Xyl*p* residue, which was confirmed from the ^1^H,^13^C-HMBC correlation from the anomeric proton at δ 4.27 (H1 of Xyl(III)) to the carbon signal at δ_C_ 80.2 (C5 of the acyl group Fa (II)). The loss of 476.3 Da in the MS^2^ spectrum corresponds to the loss of the entire acyl group with an attached pentose, as described above (Figure 2). Consequently, the structure of **Qb1** (Figure 1 top) consists of a quillaic acid moiety substituted with the trisaccharide β-d-Gal*p*-(1→2)-[ β-d-Xyl*p*-(1→3)]-β-d-Glc*p*A at C3, and the tetrasaccharide β-d-Xyl*p*-(1→4)-α-l-Rha*p*-(1→2)-[α-l-Ara*p*-(1→3)]-β-d-Fuc*p* at C28. The latter oligosaccharide is further substituted at O4 of the fucosyl unit with a glycosylated acyl group terminated by a β-d-Xyl*p*.

### 2.2. Saponin S13

Fraction B1 was further purified by semi-preparative HPLC on reverse phase column, yielding a pure saponin that was analyzed by LC-MS. This compound eluted at a retention time of 29.8 min and showed a deprotonated pseudomolecule ion [M-H]^−^ at *m*/*z* 1559.7. We previously identified this saponin in the FB extract of *Qb* using LC-MS (cf. saponin 1 in the original work [25]) and tentatively assigned it to **S13**, a saponin previously reported in *Qs* [27] (Figure 1 bottom).

A combination of 1D and 2D NMR experiments, such as multiplicity-edited ^1^H,^13^C-HSQC, ^1^H,^1^H-TOCSY, ^1^H,^13^C-HMBC, ^1^H,^13^C-H2BC, and ^1^H,^1^H-NOESY were performed and permitted the total assignment of all the ^1^H and ^13^C signals in methanol-*d4* solution. The ^1^H and ^13^C chemical shifts for the aglycone and acyl chain moieties are given in Appendix A, whereas the chemical shifts of the oligosaccharide moieties are given in Appendix A. As expected, these chemical shifts are comparable to those of **S13**, a saponin previously isolated by Nord and Kenne from *Qs* [27]. In this case, the aglycone corresponds to a 23-*O*-acetylphytolaccinic acid moiety.

The ^1^H NMR spectrum of **S13** revealed five anomeric protons that resonated as doublets at δ 5.57 (*J* = 1.5 Hz), 5.47 (*J* = 8.2 Hz), 4.55 (*J* = 7.7 Hz), 4.48 (*J* = 7.6 Hz), and 4.37 (*J* = 7.6 Hz). In the multiplicity-edited ^1^H,^13^C HSQC spectrum, these protons correlated to the carbon signals at δ_C_ 98.6, 94.8, 106.0, 105.2, and 104.9, respectively. The sugar residues were respectively assigned to α-l-Rha*p*, β-d-Fuc*p*, β-d-Gal*p*, β-d-Glc*p* and β-d-Glc*p*A. Key inter-residue correlations observed in the ^1^H,^13^C-HMBC and ^1^H,^1^H-NOESY spectra (Appendix A) allowed to confirm that the disaccharide β-d-Gal*p*-(1→2)-β-d-Glc*p*A is linked to C3 of the aglycone, and that the branched trisaccharide α-l-Rha*p*-(1→2)-[β-d-Glc*p*-(1→3)]-β-d-Fuc*p* is located at C28. The ^1^H,^13^C-HMBC spectrum showed a cross peak at δ_H_/δ_C_ 4.48/82.56 (H1 β-d-Glc*p*/C3 β-d-Fuc*p*) and the ^1^H,^1^H-NOESY spectrum showed a cross peak from the aforementioned H1 resonance to δ_H_ 4.12 (H1 β-d-Glc*p*/H3 β-d-Fuc*p*), corroborating that β-d-Glc*p* was linked to C3 of the β-d-Fuc*p* (Appendix A).The trisaccharide is substituted with two five-carbon length aliphatic acyl chains (2-methylbutanoyl acid moieties) at O3 and O4 of the fucose and rhamnose residues, respectively; furthermore, an acetyl group is located at O2 of the Rha residue. The location of the acyl groups were confirmed by the analysis of the ^1^H,^13^C-HMBC spectrum, since three-bond heteronuclear correlations could be observed form the carbonyl carbons at δ_C_ 178.6, 172.2 and 177.7 (residues Fa(I), Ac(I) and Fa(II), respectively) to the respective protons at the substitution positions: δ_H_ 5.38 (H4 β-d-Fuc*p*), 5.34 (H2 α-l-Rha*p*) and 4.98 (H3 α-l-Rha*p*), respectively.

## 3. Materials and Methods

### 3.1. Materials and Chemical Reagents

FB, an immunoadjuvant preparation obtained from the aqueous extract of *Quillaja brasiliensis* leaves, was produced by fractionation on a C18 SPE column as described previously [25,26]. **QS-21** was purchased from Desert King Chile S.A (Valparaiso, Chile). HPLC grade acetonitrile and formic acid were purchased from J. T. Baker (Phillipsburg, NJ, USA). Distilled water was purified with a Milli-Q water purification system (Millipore, Bedford, MS, USA).Thin layer chromatography (TLC) plates (precoated plates, silica gel 60, F254, 0.2 mm layer thickness) were purchased from Machery-Nagel (Duren, Germany).

### 3.2. Isolation of Saponins

#### 3.2.1. Medium-Pressure Liquid Chromatography (MPLC)

The FB (100 mg) was dissolved in 2 mL of the eluent CH_2_Cl_2_/MeOH/H_2_O/CH_3_COOH (270:139:25:1) and loaded onto a MPLC column (Buchi Borosilicat 3.3, 460 mm × 15 mm i.d., Switzerland) equipped with a precolumn (Buchi Borosilicat 3.3, 110 mm × 15 mm i.d., Switzertland), both packed with flash silica gel as the stationary phase (0.040–0.063 mm, Merck, E. Merck, Darmstadt, Germany). Chromatography was performed isocratically using the eluent described above at a flow rate of 5 mL/min and beginning to collect after passing 50 mL of the eluent through the column. The chromatographic separation of FB resulted in 46 fractions of 7 mL each, which were monitored by thin layer chromatography (TLC) for saponins. The saponin containing fractions were pooled out providing three major fractions: B1 (fractions 5–9), B2 (fractions 16–21), and B3 (fractions 26–37). The fractions were freeze-dried and kept at −20 °C until use.

#### 3.2.2. Semi-Preparative High Performance Liquid Chromatography (HPLC)

Fractions B1 and B3 were subjected to further separations using a Shimadzu LC-20AR HPLC system (Shimadzu, Kyoto, Japan) equipped with a reverse phase column (Shim-pack PREP-ODS, 250 mm × 20 mm, 5 μm, Shimadzu, Kyoto, Japan), a binary pump, and a UV-Visible detector (SPD-20AV, Shimadzu, Kyoto, Japan). Fractions were redissolved in 0.1% formic acid in water and injected into the column. Compounds were eluted with a linear gradient with 0.1% formic acid in water (A) and 0.1% formic acid in CH_3_CN (B) as the mobile phase at a flow rate of 10 mL/min. Eluent B was increased from 5–45% in 3 min, then from 45 to 53% in 20 min, then held at 100% for 6 min, and finally set back to 5% for 4 min. The detection wavelength was set at 214 nm. Purified compounds **Qb1** (2 mg, retention time at 13.5 min) and **S13** (1 mg, retention time 20.5 min) were obtained by manual collection from B3 and B1, respectively.

### 3.3. Analysis of Fractions and Purified Saponins

#### 3.3.1. Thin Layer Chromatography (TLC)

Each fraction from the MPLC system was analyzed by TLC on silica gel plates using BuOH/H_2_O/CH_3_COOH (6:2:2) as mobile phase, and anisaldehide-H_2_SO_4_/heating as detection reagent.

#### 3.3.2. Liquid Chromatography Mass Spectrometry (LC-MS)

Fractions obtained from the MPLC (B1 and B3) and HPLC systems were monitored by LC-MS. The purity of the isolated compounds (**Qb1** and **S13**) and commercial **QS-21** was also analyzed by LC-MS. The chromatography step was performed on an Ultimate 3000 RSLC systems (Dionex, Germering, Germany) coupled to a linear ion trap mass spectrometer LTQ XL from Thermo Scientific (San José, CA, USA) with an ESI interface. The instrument control and data collection were done using Xcalibur software (v3.0.63) from Thermo Scientific (San José, CA, USA) The ESI and chromatographic conditions were performed as detailed before [25,26]. The MS analysis was carried out in negative ion mode and under Full scan, Full scan MS/MS, and SIM scan modes.

#### 3.3.3. NMR Spectroscopy

Unless otherwise specified, the proton detected NMR experiments were acquired on a Bruker Avance III 500 MHz spectrometer equipped with a 5 mm Z-gradient TXI (^1^H/^13^C/^15^N) probe; the 1D ^13^C NMR spectra were recorded on a Bruker Avance III 400 MHz spectrometer equipped with a 5 mm Z-gradient BBO probe. The NMR samples were prepared by dissolving the isolated compounds (~2 for **Qb1** and ~1 mg for **S13**) and the commercial **QS-21** (~1 mg) in ~200 µL of deuterated methanol (CD_3_OD; ≥99.8 atom% D, Sigma-Aldrich, St. Louis, MO, USA) and placed in 3 mm tubes. All experiments were performed at 298 K, and the ^1^H and ^13^C chemical shifts for **Qb1** and **QS-21** are reported in ppm using the residual solvent peak as reference (δ_H_ 3.31 and δ_C_ 49.0, respectively), for **S13** using the signal of H12 and C12 of the 23-*O*-acetylphytolaccinic acid moiety as reference (δ_H_ 5.32 and δ_C_ 124.0, respectively) [27]. The assignments of the ^1^H and ^13^C resonances were obtained using 2D NMR spectra such as multiplicity-edited ^1^H,^13^C-HSQC [34], ^1^H,^1^H-TOCSY [35], and ^1^H,^13^C-HSQC-TOCSY with mixing times of 20, 40, 60, and 100 ms, ^1^H,^13^C-H2BC [36], and ^1^H,^13^C-HMBC [37]. The ^1^H,^1^H-NOESY [38] and ^1^H, ^13^C-HSQC-NOESY spectra were recorded with mixing times of 300 and 200 ms, respectively. The NMR data processing was carried out using the MestReNova (v 14.2.0) and Topspin (4.0.7) software. Considering the small amount of material recovered for **S13**, the ^1^H,^13^C-HMBC of fraction B1 was used for the analysis.

## 4. Conclusions

Herein we reported the isolation, purification, and structural characterization of two triterpene saponins from the aqueous extract of *Q. brasilliensis* leaves, including a previously undescribed isomer of the **QS-21** saponins. The chemical structure of this compound was established using a combination of mass spectrometry and 1D and 2D NMR spectroscopy. Considering the structural similarities of **Qb1** with **QS-21**, it would be expected that this compound would also display immunoadjuvant potential. In particular, this novel molecule displays three key structural features that have been previously identified as being relevant to the adjuvant activity of **QS-21:** (a) the C23 aldehyde and (b) C16 hydroxyl groups in the quillaic acid moiety, and c) the fatty acyl side chain that extends from O4 of the fucosyl residue [39]. In this regard, the isolation of **Qb1** from *Qs* leaves acquires a relevant importance, since there is an imperative need to find alternatives to the limited supply of **QS-21**, due to its low abundance in the *Qs* bark [40,41]. Further work will be necessary to determine the adjuvant activity and toxicity of **Qb1**.

## Figures and Tables

**Figure 1 molecules-27-02402-f001:**
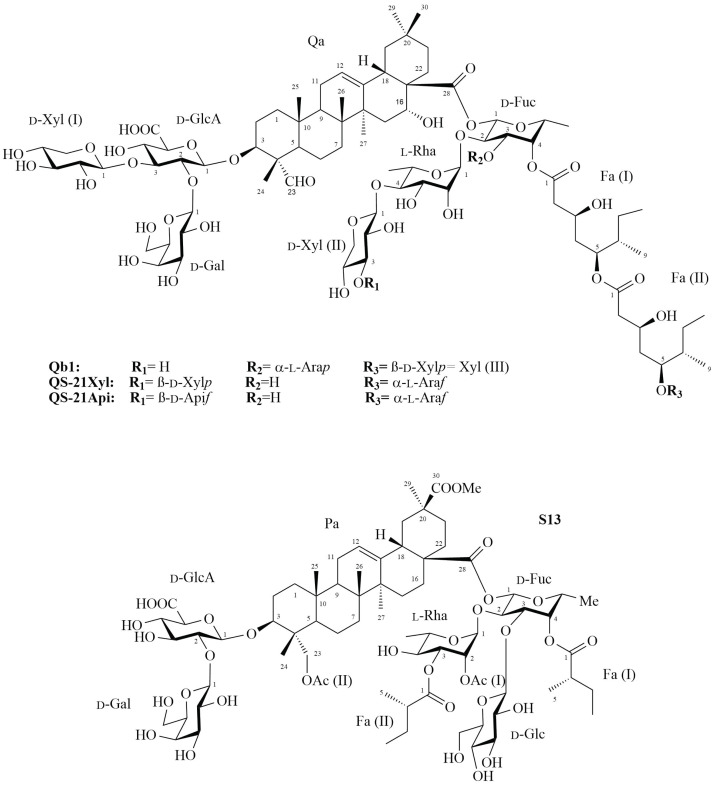
Chemical structures of saponins **Qb1** (**top**) and **S13** (**bottom**) isolated from Fraction B of *Quillaja brasiliensis*. **QS-21Xyl** and **QS-21Api** (**top**) isolated from *Quillaja saponaria*.

**Figure 2 molecules-27-02402-f002:**
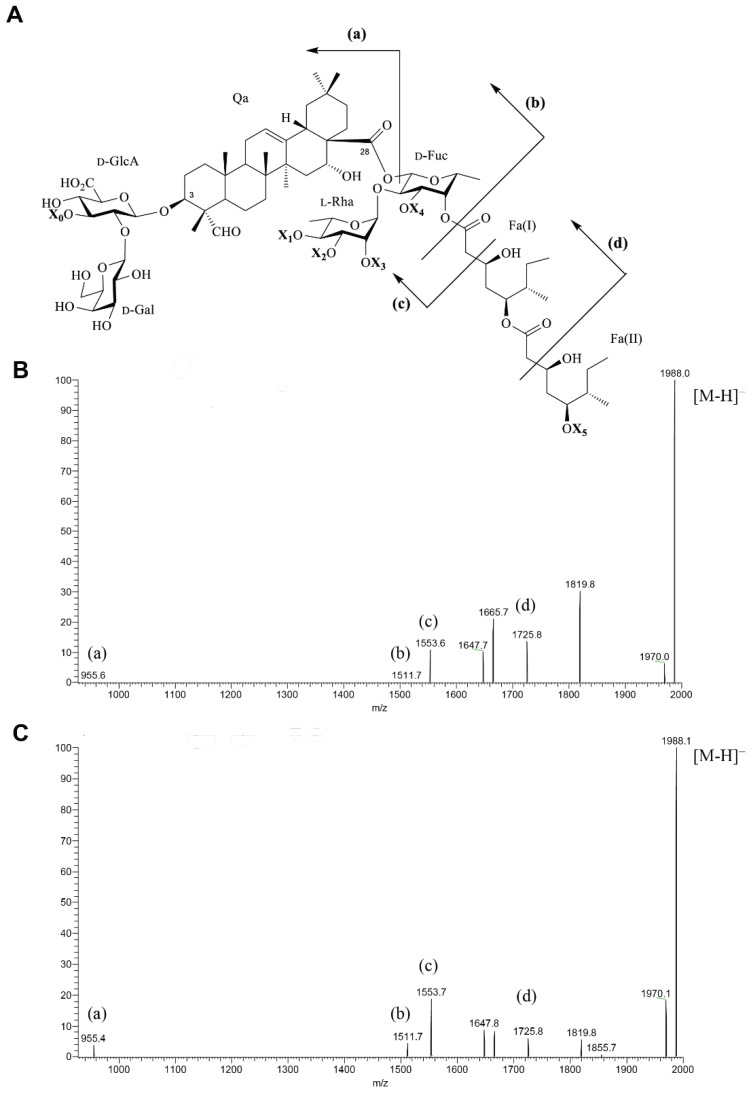
General structure of **QS-21** and isomers (**A**). Negative mode MS^2^ spectra of **Qb1** and **QS-21Xyl** (**B**,**C**, respectively) obtained from the precursor ion [M-H]^−^ at *m*/*z* 1988.0 and 1988.1, respectively. The most relevant daughter ions are assigned in the generic structure shown in A. The **X_0_** and **X_5_** substituents correspond to pentose residues in both **Qb1** and **QS-21Xyl**. Two additional pentose residues are substituting either the Rha or Fuc residues in both compounds, but their exact positions could not be determined solely by the MS^2^ data (note that in this case all the possible substitution positions are indicated as **X_1_**–**X_4_**).

**Figure 3 molecules-27-02402-f003:**
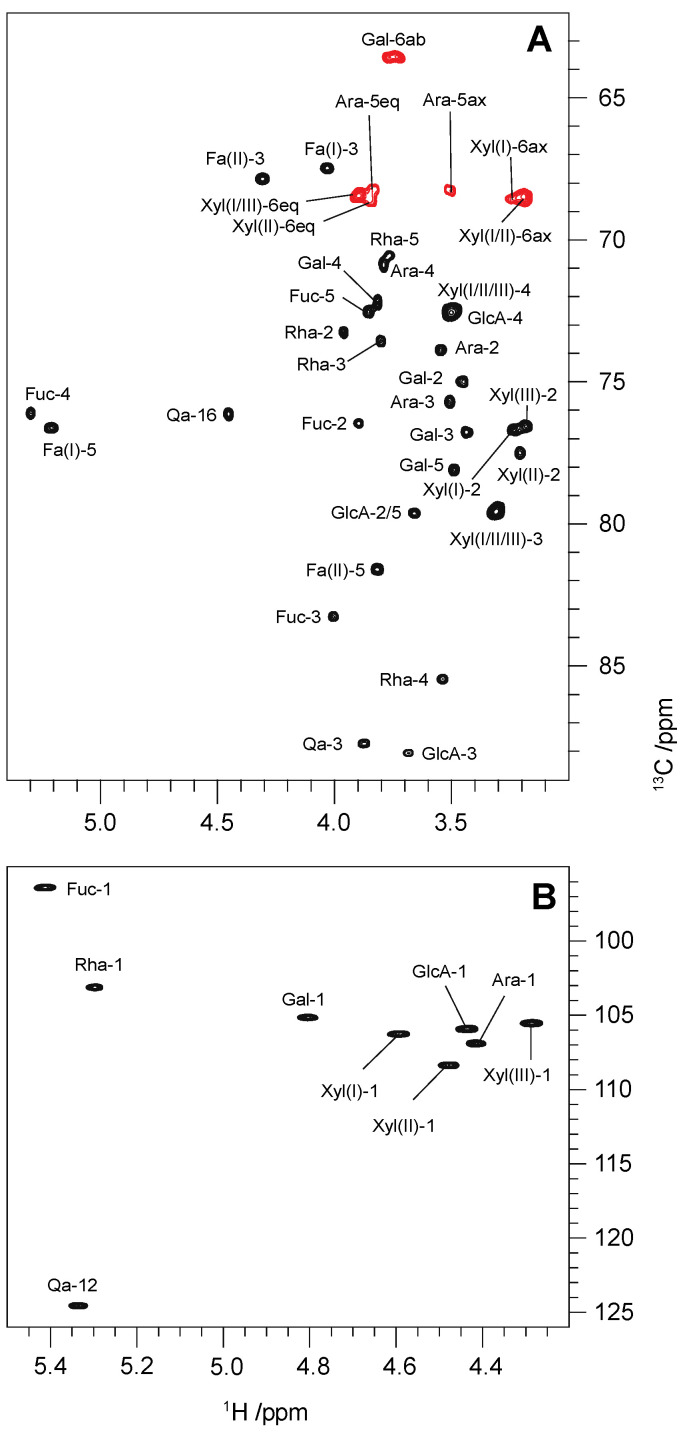
Selected regions of the multiplicity-edited ^1^H,^13^C-HSQC spectrum of **Qb1** showing the ring and hydroxymethyl groups (**A**) and the anomeric region (**B**). In the first figure (**A**), the CH_2_ groups correlations appear in red.

**Table 1 molecules-27-02402-t001:** ^1^H and ^13^C NMR chemical shifts (ppm) of the two oligosaccharide chains from **Qb1**, and inter-residue correlations from ^1^H,^1^H-NOESY and ^1^H,^13^C-HMBC spectra. The spectra were recorded in CD_3_OD at 25 °C on a Bruker Avance 500 MHz spectrometer.

Residue	Abbreviation	^1^H/^13^C						Correlation to Atom (from Anomeric Atom)	
		1	2	3	4	5	6	^1^H,^13^C-HMBC	^1^H,^1^H-NOESY
** *Qa C3-O-glycan* **									
→2,3)-β-d-Glc*p*A-(1→	GlcA	4.43 [8.6]	3.65	3.67	3.49	3.65		C3, **Qa ^(a)^**	H3, **Qa**
		104.6	~78.3	86.7	~71.2	~78.3	n.d.	H3, **Qa ^(a)^**	
β-d-Gal*p*-(1→	Gal	4.80 [7.1]	3.44	3.43	3.81	3.48	3.73, 3.76	C2, **GlcA**	H2, **GlcA**
		103.8	73.6	75.4	70.9	76.7	62.2	H2, **GlcA**	
β-d-Xyl*p*-(1→	Xyl(I)	4.59 [7.7]	3.23	3.30	3.49	3.18, 3.89		C3, **GlcA**	H3, **GlcA**
		104.9	75.3	~78.2	~71.2	~67.1		H3, **GlcA**	
** *Qa C28-O-glycan* **									
→2,3,4)-β-d-Fuc*p*-(1→	Fuc	5.41 [8.1]	3.89	4.00	5.29	3.85	1.08	C28, **Qa ^(a)^**	
		95.1 {164}	75.1	81.9	74.7	71.2	16.9		
→4)-α-l-Rha*p*-(1→	Rha	5.29 [1.9]	3.96	3.80	3.56	3.77	1.30	C2, **Fuc**	H2, **Fuc**
		101.8 {172}	71.9	72.2	84.1	69.2	18.5	H2, **Fuc**	
β-d-Xyl*p*-(1→	Xyl(II)	4.47 [7.7]	3.20	3.30	3.49	3.18, 3.89		C4, **Rha**	H4, **Rha**
		107.0	76.1	~78.2	~71.2	~67.3		H4, **Rha**	
β-d-Xyl*p*-(1→	Xyl(III)	4.27 [7.7]	3.18	3.30	3.49	3.18, 3.89		C5, **Fa(II) ^(a)^**	
		104.2	75.2	~78.2	~71.2	~67.1		H5, **Fa(II) ^(a)^**	
α-l-Ara*p*-(1→	Ara	4.41 [6.7]	3.54	3.51	3.78	3.49, 3.84		C3, **Fuc*p***	H3, **Fuc**
		105.6	72.5	74.3	69.5	66.9		H3, **Fuc*p***	

^3^*J*_H1,H2_ values are given in hertz in square brakets and ^1^*J*_C1,H1_ are given in braces. ^(a)^ Chemical shifts of these atoms are shown in Appendix A.

## Data Availability

The data are available within the article and its Appendix A.

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
