# Peer review of "Structure Elucidation of Triterpenoid Saponins Found in an Immunoadjuvant Preparation of Quillaja brasiliensis Using Mass Spectrometry and 1H and 13C NMR Spectroscopy"

_molecules, 2022, doi:10.3390/molecules27082402_

Round 1
Reviewer 1 Report
Dear Authors,
The presented manuscript address the isolation and structure elucidation of two triterpenoid saponins that probably will be used in vaccines. It has scientific soundness. Nevertheless, some points require modification.
In the title, it is suggested to mention mass spectrometry besides NMR.
In all the chemical structures, it is strongly advised to add complete numbering to the atoms. Also, make different numbering for sugars conjugates as well as sidechains. (e.g. 1, 1',1'').
You have also to state the abbreviations in their first appearance as MPLC, HPLC and so on.
line 94 correspond --> corresponded
line 123 please remove the extra word 'point'
In the materials part, please add the companies names and their addresses for the used columns.
Thank you
Author Response
We greatly appreciate all the valuable comments and suggestions that helped us to improve the quality of our manuscript. We considered all the suggestions and comments annotated by the Reviewer, as are detailed below.
-The presented manuscript address the isolation and structure elucidation of two triterpenoid saponins that probably will be used in vaccines. It has scientific soundness.
We thank the reviewer for the comment.
-In the title, it is suggested to mention mass spectrometry besides NMR.
As suggested, we have rewritten the title mentioning mass spectrometry. The new title is “Structure elucidation of triterpenoid saponins found in an immunoadjuvant preparation of Quillaja brasiliensis using mass spectrometry and 1H and 13C NMR spectroscopy”.
-In all the chemical structures, it is strongly advised to add complete numbering to the atoms. Also, make different numbering for sugars conjugates as well as sidechains. (e.g. 1, 1', 1'').
We prefer to avoid prime, double prime, triple prime, etc. numbering since the figure will become too crowded. Each atom is identified by the residue abbreviation (i.e. GlcA, Gal, Xyl(I), Fuc, Rha, Ara, Xyl(I), Xyl(II), Xyl(III) in the case of monosaccharide residues, Fa(I) and Fa(II) in the case of fatty acyl chains, and Qa in the case of the quillaic acid residue) and the atom position in the corresponding residue.
To facilitate the reading of the manuscript we have added numbers to the substitution positions of the monosaccharide residues in Figure 1.
-You have also to state the abbreviations in their first appearance as MPLC, HPLC and so on.
Done.
-line 94 correspond --> corresponded
Done. We have replaced “correspond” with “corresponded”.
-line 123 please remove the extra word 'point'
Done.
-In the materials part, please add the companies names and their addresses for the used columns.
Done. We have replaced “equipped with a reverse phase column (Shim-pack C18, 250 × 20 mm, 5 μm)” with “equipped with a reverse phase column (Shim-pack PREP-ODS, 250 × 20 mm, 5 μm, Shimadzu, Kyoto, Japan)”. Also we have replaced in the same paragraph “UV-Visible detector (SPD-20AV, Shimadzu)” with “UV-Visible detector (SPD-20AV, Shimadzu, Kyoto, Japan)”.
-Moderate English changes required.
We have improved the wording in English. These changes were light blue in the revised manuscript.
Again, we greatly appreciate all the valuable comments and suggestions that helped us to improve the quality of our manuscript.
Reviewer 2 Report
The manuscript describes the isolation, structure elucidation, and complete 1H and 13C assignment of triterpenoid saponins found in preparation of immunoadjuvant of Quillaja brasiliensis.
The introduction and references are appropriate, the results are suitably presented, the discussion is relevant, experimental methods and procedures are well described, compounds are adequately characterized, and the conclusions are clear.
The authors have performed a thorough structural elucidation based on a series of NMR techniques and by correlation of relevant NMR data. Although it is quite difficult to check and verify such complex sets of spectral data, it seems to me that the authors have done the structural assignments correctly. Nevertheless, additional confirmation of the key-structures by X-ray analysis would wipe away any ambiguity about the correctness of structure determination.
So, I recommend publication of the manuscript in the present form in Molecules journal.
Author Response
Responses to Reviewer 2
We greatly appreciate all the valuable comments and suggestions that helped us to improve the quality of our manuscript. We considered all the suggestions and comments annotated by the Reviewer, as are detailed below.
- The manuscript describes the isolation, structure elucidation, and complete 1H and 13C assignment of triterpenoid saponins found in preparation of immunoadjuvant of Quillaja brasiliensis. The introduction and references are appropriate, the results are suitably presented, the discussion is relevant, experimental methods and procedures are well described, compounds are adequately characterized, and the conclusions are clear. The authors have performed a thorough structural elucidation based on a series of NMR techniques and by correlation of relevant NMR data. Although it is quite difficult to check and verify such complex sets of spectral data, it seems to me that the authors have done the structural assignments correctly. Nevertheless, additional confirmation of the key-structures by X-ray analysis would wipe away any ambiguity about the correctness of structure determination. So, I recommend publication of the manuscript in the present form in Molecules journal.
We thank the reviewer for the comment. We were pleased that the paper was appreciated.
The isolated saponins were obtained as amorphous solids and in limited quantity. The crystallization process is very complex, therefore confirmation of the structure by x-ray analysis was not an option.